# Improved Unsupervised Chinese Word Segmentation Using Pre-trained Knowledge and Pseudo-labeling Transfer

**Hsiu-Wen Li**[*], **Ying-Jia Lin**[*], **Yi-Ting Li, Chun Yi Lin , Hung-Yu Kao**

Intelligent Knowledge Management Lab
Department of Computer Science and Information Engineering
National Cheng Kung University
Tainan, Taiwan
{allen52110, yingjia.lin.public, yitingli.public}@gmail.com,
ne6101050@gs.ncku.edu.tw, hykao@mail.ncku.edu.tw

## Abstract

Unsupervised Chinese word segmentation (UCWS) has made progress by incorporating linguistic knowledge from pre-trained language models using parameter-free probing techniques. However, such approaches suffer from increased training time due to the need for multiple inferences using a pre-trained language model to perform word segmentation. This work introduces a novel way to enhance UCWS performance while maintaining training efficiency. Our proposed method integrates the segmentation signal from the unsupervised segmental language model to the pre-trained BERT classifier under a pseudo-labeling framework. Experimental results demonstrate that our approach achieves state-of-the-art performance on the seven out of eight UCWS tasks while considerably reducing the training time compared to previous approaches.

## 1 Introduction

Word segmentation is critical in natural language processing (NLP) tasks. Different from English, Chinese text does not show explicit delimiters between words (e.g., whitespaces), making Chinese word segmentation (CWS) a challenging task. Traditional unsupervised Chinese word segmentation (UCWS) approaches include rule-based and statistical methods (Chang and Lin, 2003; Tseng et al., 2005; Low et al., 2005; Mochihashi et al., 2009). In recent years, neural approaches based on word embeddings and recurrent neural networks have been studied for UCWS. Sun and Deng (2018) proposed Segmental Language Model (SLM) with the pre-trained word embeddings (Mikolov et al., 2013) and the LSTM-based language model (Hochreiter and Schmidhuber, 1997). By calculating the probabilities between words in a fixed-length segmentation size, SLM performs better than the traditional approaches on the UCWS tasks.

---

[*]The first two authors contributed equally.

Recently, as large-scale pre-trained language models have become mainstream solutions for several NLP tasks, Li et al. (2022) successfully applied BERT (Devlin et al., 2018) to UCWS and demonstrated state-of-the-art (SOTA) performance by incorporating Perturb Masking (Wu et al., 2020) and the self-training loops into the BERT classifier. However, Perturb Masking is computationally inefficient due to the requirement of performing at least twice BERT forward passes for each token in left-to-right order (Li et al., 2022). In other words, for a sequence with a length of $N$, the additional training time complexity using Perturb Masking (Wu et al., 2020) will become $2N$, which significantly increases the training time of fine-tuning BERT for UCWS.

This paper introduces a simple unsupervised training framework to leverage the pre-trained BERT (Devlin et al., 2018) for UCWS efficiently. Following Xue (2003), we view the CWS task as a sequence tagging problem. To make the BERT model learn how to segment words with its implicit pre-trained knowledge, we propose a pseudo-labeling approach to fine-tune BERT with the pseudo-labels generated by an unsupervised segment model. Our experiments demonstrate that the proposed method brings substantial performance gains compared to the previous studies (Sun and Deng, 2018; Downey et al., 2022; Li et al., 2022). In addition, the proposed method provides an 80% decrease in training time than the existing SOTA method (Li et al., 2022), which also utilized the pre-trained language model for UCWS.

## 2 Method

There are two modules in the proposed framework: the segment model and the classifier, which we show in Figure 1. The segment model produces the text segmentation results, which serve as pseudo-labels. The classifier, which is a BERT-based classification model, then uses these labels to learn

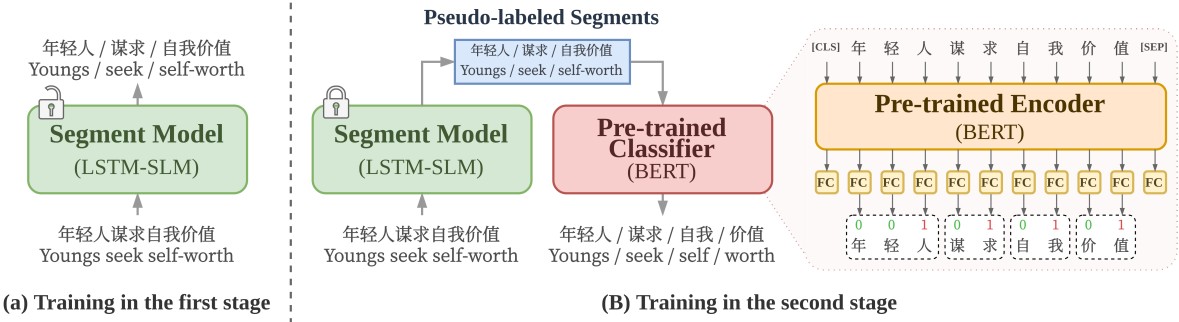

Figure 1: The proposed training framework for unsupervised Chinese word segmentation.

how to separate the words in a sequence. In other words, the segment model acts as a teacher for the classifier. The following sections first introduce the details of the two modules and then describe the training process in our proposed framework.

## 2.1 Segment Model

In our approach, we employ the Segment Language Model (SLM) proposed by Sun and Deng (2018) as our segment model for providing pseudo-labels in our proposed framework. SLM is a language model which segments a sequence according to the probability of generating <eos> (*end of a segment*) as the next character. For example, given a sequence $\{x_1, x_2, x_3\}$, two segments $\{x_1, x_2\}$ and $\{x_3\}$ can be obtained via SLM if the probability of generating <eos> after $x_2$ is higher than that after $x_3$. We omit an exhaustive unsupervised training process of SLM (Sun and Deng, 2018) and refer readers to the original paper for more details.

## 2.2 Classifier

We follow Xue (2003) to treat CWS as a sequence tagging task. As the illustration in Figure 1 (b), we add fully-connected layers (FC) on top of the BERT (Devlin et al., 2018) model as our classifier:

$$p_{ij} = \text{softmax}(W \cdot h_{ij} + b),$$
$$\forall i \in [1, N], \forall j \in [1, T_i],$$

where $p_{ij}$ is the probability of the $j$-th token in the $i$-th sequence, $N$ is the number of examples in a dataset, and $T_i$ is the length of the $i$-th sequence. $W \in \mathbb{R}^{d \times k}$ and $b \in \mathbb{R}^k$ are trainable parameters for $k$ output tagging labels, and $h_{ij} \in \mathbb{R}^d$ is the output hidden state of BERT at the $j$-th token with $d$ dimensions.

## 2.3 Training Framework

Our training framework is composed of two stages, as illustrated in Figure 1. In the first stage, we follow Sun and Deng (2018) to train the segment model. We then perform pseudo-labeling and create high-quality word segments with the segment model. In the second stage, we use these pseudo-labeled segments to train the classifier with cross-entropy loss:

$$\mathcal{L} = -\sum_{i=1}^{N} \sum_{j=1}^{T_i} y_{ij} \log(p_{ij})$$

where $y_{ij}$ is the pseudo-label of the $j$-th token in the $i$-th sequence. We adopt the tagging schema with binary tagging labels ($k = 2$), where "1" represents "*segment from the next character*" and "0" indicates "*do not segment*." The results with other tagging schemas are included in Appendix A.1.

Our goal is to provide the pre-trained classifier with pseudo-labels as training data, allowing it to utilize the knowledge acquired during its pre-training stage to learn further and improve its performance on Chinese word segmentation.

## 3 Experiments

### 3.1 Datasets

We conduct experiments on the eight widely used datasets, including AS, CityU, MSR and PKU from the SIGHAN 2005 Bakeoff (Emerson, 2005), SXU from the SIGHAN 2008 Bakeoff (Jin and Chen, 2008), CNC and CTB from the Penn Treebank (Xue et al., 2005), and UD from the CoNLL 2018 Shared Task (Zeman et al., 2018). We use the evaluation script provided by SIGHAN 2005 (Emerson, 2005) and report the results in F1 scores.

| Model | AS | CityU | MSR | PKU | CNC | CTB | SXU | UDC | Avg. |
|---|---|---|---|---|---|---|---|---|---|
| SLM-4 (Sun and Deng, 2018) | 79.7 | 80.3 | 79.6 | 79.6 | 79.6 | 78.4 | 81.0 | 71.6 | 78.7 |
| MSLM (Downey et al., 2022) | 40.3 | 67.4 | 71.2 | 67.1 | 48.9 | 63.4 | 64.3 | 44.3 | 58.3 |
| Bert-Circular (Li et al., 2022) | 79.0 | 82.6 | **82.9** | 83.7 | 81.6 | 81.0 | 83.6 | 82.8 | 82.2 |
| Ours | **86.4** $_{(0.1)}$ | **85.7** $_{(0.5)}$ | 82.3 $_{(0.2)}$ | **84.3** $_{(0.3)}$ | **85.6** $_{(0.1)}$ | **84.2** $_{(0.3)}$ | **86.1** $_{(0.2)}$ | **83.8** $_{(0.4)}$ | **84.8** $_{(0.3)}$ |

Table 1: Performance in F1-score (%) on the eight CWS datasets. The scores of baselines are reported as the best one from 5 runs. In contrast, our results display the average score of 5 experiments for each dataset, with the standard deviation indicated as a subscript.

| Model | Original | Implemented | Time |
|---|---|---|---|
| SLM-4 | 79.2 | 79.6 | 31m |
| MSLM | 62.9 | 67.1 | 119m |
| BERT-circular | 84.1 | 83.7 | 186m |
| Ours | - | **84.3** | **38m** |

Table 2: Training time comparison (in minutes) on the PKU dataset. The underlined scores are taken from the original papers.

| | AS | CityU | MSR | PKU | CNC |
|---|---|---|---|---|---|
| **Proposed** | 86.4 | 85.6 | 82.4 | 84.3 | 85.6 |
| Proposed + ST | 86.5 | 85.6 | 82.6 | 84.8 | 85.8 |

Table 3: Performance in F1-score (%) of using self-training in the proposed framework.

## 3.2 Implementation Details

We follow Sun and Deng (2018) to replace the continuous English characters with the special token <eng>, digits with the token <num>, and punctuations with the token <punc>. We use the CBOW model (Mikolov et al., 2013) which has been pre-trained on the Chinese Gigaword Corpus (LDC2011T13) to acquire word representations for our segment model. The encoder and the decoder of the segment model are composed of a one-layer LSTM and a two-layer LSTM. For the classifier, we use the pre-trained Chinese BERT$_{BASE}$ model[1]. We use Adam (Kingma and Ba, 2015) with learning rates of 5e-3 for the segment model and [1e-5, 5e-5] for the classifier. We train the segment model for 6,000 steps for each dataset, including the initial 800 steps for linear learning rate warm-up. The classifier is trained for 1,000 steps on each task with early stopping. More training details and training progress of our approach without using early stopping can be found at Appendix A.2 and A.3.

## 3.3 Results

Table 1 shows the results for UCWS on the eight datasets. Our baselines include SLM-4 (Sun and Deng, 2018), MSLM (Downey et al., 2022), and BERT-circular (Li et al., 2022). We demonstrate that the proposed approach outperforms the baselines for UCWS. Although our approach is slightly worse than the existing SOTA method (Li et al., 2022) on the MSR dataset (Table 1), we still ob-

---

[1] https://huggingface.co/bert-base-chinese.

serve the substantial performance gains for the remaining seven Chinese word segmentation tasks. Note that the methods we compare in Table 1 did not include complete results on the eight tasks in their original papers, so the remaining scores were obtained by our own implementations. We compare our re-implementation results with the baseline methods in Appendix A.4 on the datasets presented in their original papers.

## 3.4 Training Speed Comparison

As previously mentioned, our work aims at simplifying the framework of BERT-circular (Li et al., 2022) to reduce the training time for UCWS. Here, we test the training speed to compare the proposed method and the baselines using a single GPU of RTX 3090. Table 2 shows that the proposed method takes only 20% of the training time of BERT-circular but performs better on the PKU dataset. In addition, our approach is better than MSLM (Downey et al., 2022) in both training speed and model performance. However, compared to the SLM (Sun and Deng, 2018), our method needs a slightly longer training time due to the training of the classifier with pseudo-labeling.

## 4 Analysis

### 4.1 Use Self-Training?

Li et al. (2022) include self-training in their framework and improve the model performance. Thus, this section reveals if our training framework also benefits from the self-training technique. We first follow the proposed approach to train the classifier. Then we iteratively train the segment model and the classifier with the pseudo-labels from each of the two modules until early-stopping. Table 3 shows

| Model | Segmentation Example |
|---|---|
| Gold | 大连 / 冰山 / 自行车 / 俱乐部 / 奖励 / 奥运 / 功臣 |
| SLM | 大连 / 冰山 / 自行 / 车 / 俱乐部 / 奖励 / 奥运 / 功臣 |
| BERT-circular | 大连 / 冰山 / 自行车 / 俱乐 / 部 / 奖励 / 奥运功臣 |
| Ours | 大连 / 冰山 / 自行车 / 俱乐部 / 奖励 / 奥运 / 功臣 |
| Gold | 特拉维夫 / 北部 / 约 / 30 / 公里 / 的 / 海滨 / 城市 / 内坦亚 |
| SLM | 特拉 / 维夫 / 北部 / 约 / 30 / 公里 / 的海滨 / 城市 / 内坦亚 |
| BERT-circular | 特拉 / 维夫 / 北部 / 约 / 30 / 公里 / 的 / 海滨 / 城市 / 内坦亚 |
| Ours | 特拉维夫 / 北部 / 约 / 30 / 公里 / 的 / 海滨 / 城市 / 内坦亚 |
| Gold | 哈尔滨市 / 还 / 联袂 / 推出 / 兆麟 / 公园 / 冰灯 / 艺术 / 博览会 |
| SLM | 哈尔滨 / 市 / 还 / 联袂 / 推出 / 兆麟 / 公园 / 冰 / 灯 / 艺术 / 博览会 |
| BERT-circular | 哈尔 / 滨 / 市 / 还 / 联袂 / 推出 / 兆麟 / 公园 / 冰灯 / 艺术 / 博览 / 会 |
| Ours | 哈尔滨 / 市 / 还 / 联袂 / 推出 / 兆麟 / 公园 / 冰灯 / 艺术 / 博览会 |

Table 4: Segmentation examples from the PKU dataset.

that self-training (ST) brings a subtle improvement to our proposed framework. We argue that a filtering strategy for low-confidence examples should be combined with self-training, which will be further studied in our future work.

## 4.2 Segmentation Examples

Table 4 provides three examples of CWS. Across these examples, both SLM and BERT-circular exhibit a mixture of correct and incorrect word segmentation results. For instance, in the first example, "自行车" (bicycle) is incorrectly segmented as "自行/车" (self / bicycle) by SLM, while BERT-circular segments it correctly. Conversely, "俱乐部" (club) is wrongly segmented by BERT-circular, while SLM is correct. Notably, our model excels in correctly segmenting proper nouns, as seen with "特拉维夫" (Tel Aviv), where both SLM and BERT-circular falter. However, our model does tend to encounter challenges with complex terms, such as the combination of a proper noun and a standard term, exemplified by "哈尔滨" + "市" (Harbin city). Despite this, our method is able to leverage the insights from both SLM and BERT, achieving accurate segmentation in numerous instances. See Table 9 in Appendix for more examples.

We also discover that BERT-circular shows unsatisfactory results when segmenting words longer than two characters, such as "俱乐部" (club) and "奥运功臣" (Olympic hero). Therefore, we analyze the relationship between performance and segmentation lengths in the next section.

## 4.3 Comparison of Model Performance on Different Segmentation Lengths.

Figure 2 shows the performance comparison for different segmentation lengths of the PKU dataset

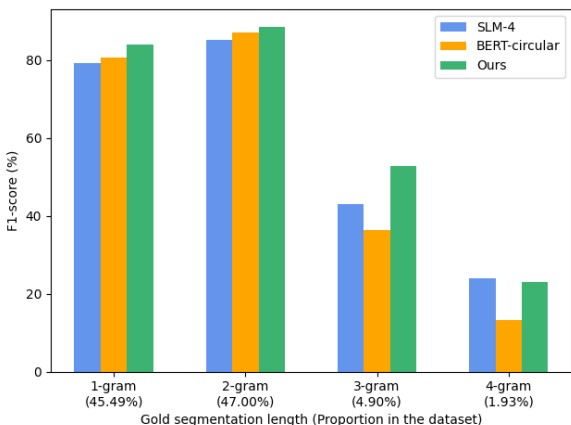

Figure 2: Comparison of model performance in F1-score on the different segmentation lengths using the PKU dataset. The x-axis shows the gold segmentation lengths, with the proportion denoted in parentheses.

in 1-gram, 2-gram, 3-gram, and 4-gram. Although BERT-circular (Li et al., 2022) uses the pre-trained BERT model (Devlin et al., 2018) in its framework, it fails to provide satisfying results for longer segments (3-gram and 4-gram). In contrast, our method performs well in most settings and shows competitive results for 4-gram segmentations compared with SLM (Sun and Deng, 2018). The reason for the results of BERT-circular may come from Perturb Masking (Wu et al., 2020), which measures the relationship between every two left-to-right tokens and relies on the similarity between the two representations. The segmentation results of BERT-circular may be less accurate when computing similarities multiple times for longer segments.

## 5 Related work

Sun and Deng (2018) first built the LSTM-based segmental language model using neural approaches

for modeling Chinese fragments without labels. To perform unsupervised Chinese word segmentation, they leveraged dynamic programming to find the optimal possible segmentations based on the word probabilities of the LSTM-based language model. Due to the invention of the self-attention mechanism (Vaswani et al., 2017), Downey et al. (2022) carefully designed a masking strategy for replacing the LSTM-based architecture of the SLM with Transformer (Vaswani et al., 2017). Wang and Zheng (2022) integrated the forward and backward segmentations instead of only using forward information as in previous work.

In terms of pre-trained semantic knowledge usage, Wu et al. (2020) developed the Perturbed Masking probing to assess the relations between tokens in a sentence from the masked language model of BERT (Devlin et al., 2018). Based on the probing approach, Li et al. (2022) proposed a self-training manner that makes the classifier learn to divide word units from the perturbed segmentation and achieved SOTA performance for UCWS.

## 6 Conclusion

This work presents an improved training framework for unsupervised Chinese word segmentation (UCWS). Our framework leverages the pseudo-labeling approach to bridge the two-stage training of the LSTM-based segment model and the pre-trained BERT-Chinese classifier. The experiments show that the proposed framework outperforms the previous UCWS methods. In addition, without using Perturb Masking (Wu et al., 2020) and self-training (Li et al., 2022), our framework significantly reduces training time compared with the SOTA approach (Li et al., 2022). Our code is available at `https://github.com/IKMLab/ImprovedUCWS-KnowledgeTransfer`.

## 7 Limitations

Our segment model (Sun and Deng, 2018) requires the pre-trained word embedding (Mikolov et al., 2013) as initialization of the embedding layer. Random initialization of the embedding layer might lead to slow convergence of the segment model.

## 8 Acknowledgements

This work was supported by the National Science and Technology Council, Taiwan, under Grant NSTC 112-2223-E-006-009. We would like to thank the reviewers for their insightful feedback. We also thank Dr. Reinald Adrian Pugoy for checking our manuscript.

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

# A Appendix

## A.1 Additional results for different tagging schemas

We show the two additional tagging schemas, BI (*beginning*, *inside*) and BMES (*beginning*, *middle*, *end*, *single*), for evaluating the performance of the proposed method in Table 5. The results show minor differences in the "01" tagging schema and the other two for the eight unsupervised Chinese word segmentation (UCWS) tasks.

## A.2 Training Details

Table 6 shows the information of the train and test set splits. In our training framework, we use the train set in each CWS dataset for unsupervised training of the segment model. Afterward, we infer each example in a test set with the fixed segment model to acquire the segmentation results as pseudo-labels for training the BERT-based classifier. For evaluations of UCWS, we use the trained classifier to predict the examples in test sets, which follows the approach of Li et al. (2022).

## A.3 Performance without Early Stopping

The line chart shown in Figure 3 illustrates the training outcomes of our framework in the second stage, which did not use early stopping. The curves in Figure 3 show an initial upward trend, followed by a subsequent decrease before stabilizing. In the second stage, we train the BERT-based classifier using pseudo-labels. We believe that BERT's pre-trained knowledge (Devlin et al., 2018) can improve word segmentation quality in the initial training phases. Therefore, we include the early stopping mechanism to ensure the enhancement of word segmentation. Without early stopping, the BERT-based classifier would be influenced by the distribution of pseudo-labels generated by the SLM (Sun and Deng, 2018).

## A.4 Re-implementation Results

We evaluate the proposed method on the eight CWS datasets (Table 1). However, not all of them are reported in the original publications of the compared baseline methods. In order to validate the re-implementation results, Table 7 compares the scores obtained by our implementations on the datasets presented in the original publications. According to the results, our scores of SLM (Sun and Deng, 2018) and MSLM (Downey et al., 2022) are close to the ones reported in their original papers.

However, we cannot reproduce the scores consistent with those presented in the BERT-circular paper (Li et al., 2022) using their publicly released code [2] and the hyperparameters. We discovered that the results of BERT-circular are not deterministic due to the lack of randomness control in their code. Indeed, as the main contributions stated in their paper (Li et al., 2022), BERT-circular shows much better performance than SLM (Sun and Deng, 2018), which is consistent with our results and can also be observed in Tables 1 and 7. Additional experiments can be found in Table 8, which showcases five experiments conducted for each approach on every CWS dataset.

---

[2] https://github.com/liweitj47/BERT_unsupervised_word_segmentation.

| Tags | AS | CityU | MSR | PKU | CNC | CTB | SXU | UDC | Avg. |
|---|---|---|---|---|---|---|---|---|---|
| 01 (Reported) | 86.5 | 85.3 | 82.2 | 84.1 | 85.6 | 84.1 | 86.0 | 83.5 | 84.7 |
| BI | 85.9 | 85.3 | 81.6 | 83.9 | 85.3 | 83.2 | 85.2 | 80.4 | 83.9 |
| BMES | 85.5 | 84.7 | 82.0 | 83.7 | 85.2 | 84.2 | 86.9 | 82.7 | 84.4 |

Table 5: Performance in F1-score (%) on the eight datasets using different tagging schemas.

| Data Split | AS | CityU | MSR | PKU | CNC | CTB | SXU | UDC |
|---|---|---|---|---|---|---|---|---|
| Training set | 708k | 53k | 86k | 19k | 207k | 24k | 17k | 39k |
| Test set | 14k | 1.4k | 3.9k | 1.9k | 25k | 1.9k | 3.6k | 0.5k |

Table 6: Number of examples in each Chinese word segmentation dataset.

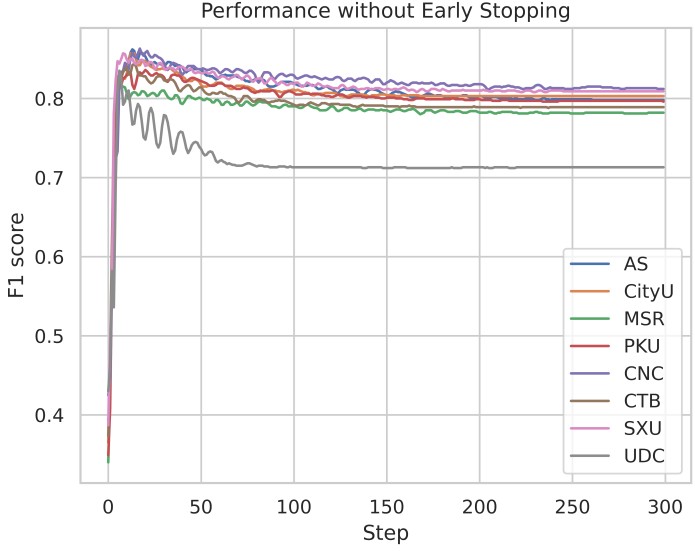

Figure 3: Training progress of the BERT-based classifier in the second stage without the use of early stopping.

| Model | AS | CityU | MSR | PKU |
|---|---|---|---|---|
| SLM-4 (Sun and Deng, 2018) | 79.8 | 79.7 | 79.0 | 79.2 |
| SLM-4* | 79.7 | 80.3 | 79.6 | 79.6 |
| MSLM (Downey et al., 2022) | - | - | - | 62.9 |
| MSLM* | - | - | - | 67.1 |
| BERT-circular (Li et al., 2022) | - | - | 83.0 | 84.1 |
| BERT-circular* | - | - | 82.9 | 83.7 |

Table 7: Comparison of F1-score (%) on the same datasets included in the original publications of the baseline methods. Our reimplementations are marked with an asterisk (*) where the score is the best one from 5 experiments.

| Model | | AS | CityU | MSR | PKU | CNC | CTB | SXU | UDC | Avg. |
|---|---|---|---|---|---|---|---|---|---|---|
| SLM-4 (Sun and Deng, 2018) | Run 1 | 79.6 | 80.1 | 79.4 | 79.4 | 78.8 | 78.3 | 80.8 | 71.6 | 78.5 |
| | Run 2 | 79.7 | 80.2 | 79.2 | 79.4 | 79.6 | 78.1 | 81.0 | 71.1 | 78.5 |
| | Run 3 | 79.7 | 80.2 | 79.6 | 79.5 | 78.7 | 78.4 | 80.9 | 71.2 | 78.5 |
| | Run 4 | 79.6 | 80.3 | 79.3 | 79.4 | 78.8 | 78.3 | 80.9 | 70.9 | 78.4 |
| | Run 5 | 79.7 | 80.3 | 79.6 | 79.6 | 78.9 | 78.1 | 80.8 | 70.9 | 78.5 |
| | Avg. | 79.7 | 80.2 | 79.4 | 79.5 | 79.0 | 78.2 | 80.9 | 71.1 | 78.5 |
| | Std. | 0.05 | 0.08 | 0.18 | 0.09 | 0.36 | 0.13 | 0.08 | 0.29 | 0.16 |
| MSLM (Downey et al., 2022) | Run 1 | 40.3 | 67.2 | 71.2 | 59.6 | 48.9 | 59.6 | 64.3 | 30.2 | 55.2 |
| | Run 2 | 39.0 | 66.6 | 58.8 | 64.2 | 45.5 | 63.4 | 60.3 | 28.1 | 53.2 |
| | Run 3 | 39.4 | 66.0 | 57.9 | 56.8 | 45.5 | 55.2 | 61.1 | 33.3 | 51.9 |
| | Run 4 | 38.8 | 67.4 | 68.3 | 62.0 | 43.7 | 43.0 | 59.4 | 32.4 | 51.9 |
| | Run 5 | 39.9 | 66.8 | 61.8 | 67.1 | 47.9 | 51.4 | 61.2 | 44.3 | 55.1 |
| | Avg. | 39.5 | 66.8 | 63.6 | 61.9 | 46.3 | 54.5 | 61.3 | 33.7 | 53.5 |
| | Std. | 0.62 | 0.55 | 5.89 | 3.99 | 2.08 | 7.87 | 1.85 | 6.28 | 3.64 |
| BERT-circular (Li et al., 2022) | Run 1 | 76.2 | 80.1 | 80.2 | 79.8 | 79.8 | 79.4 | 82.5 | 81.6 | 80.0 |
| | Run 2 | 79.0 | 80.2 | 80.8 | 83.7 | 77.8 | 81.0 | 83.6 | 82.8 | 81.1 |
| | Run 3 | 76.6 | 79.5 | 78.2 | 82.4 | 81.2 | 80.8 | 81.6 | 82.1 | 80.3 |
| | Run 4 | 74.4 | 82.6 | 77.9 | 83.2 | 77.2 | 79.4 | 81.8 | 82.5 | 79.9 |
| | Run 5 | 78.2 | 78.4 | 82.9 | 79.6 | 81.6 | 78.3 | 83.0 | 81.7 | 80.5 |
| | Avg. | 76.9 | 80.2 | 80.0 | 81.7 | 79.5 | 79.8 | 82.5 | 82.1 | 80.3 |
| | Std. | 1.8 | 1.54 | 2.05 | 1.92 | 1.97 | 1.12 | 0.83 | 0.51 | 1.5 |
| Ours | Run 1 | 86.4 | 85.2 | 82.1 | 84.1 | 85.5 | 84.0 | 85.9 | 83.4 | 84.6 |
| | Run 2 | 86.5 | 85.3 | 82.2 | 84.1 | 85.6 | 84.1 | 86.0 | 83.5 | 84.7 |
| | Run 3 | 86.4 | 86.1 | 82.5 | 84.3 | 85.7 | 84.3 | 86.1 | 84.2 | 85.0 |
| | Run 4 | 86.6 | 86.2 | 82.6 | 84.7 | 85.7 | 84.7 | 86.4 | 84.3 | 85.2 |
| | Run 5 | 86.3 | 85.6 | 82.4 | 84.1 | 85.7 | 84.1 | 86.0 | 83.6 | 84.7 |
| | Avg. | 86.4 | 85.7 | 82.4 | 84.3 | 85.6 | 84.2 | 86.1 | 83.8 | 84.8 |
| | Std. | 0.11 | 0.45 | 0.21 | 0.26 | 0.09 | 0.28 | 0.19 | 0.42 | 0.25 |

Table 8: The F1 score (%) performance of five experiments on each of the eight CWS datasets.

| Model | Segmentation Example |
|---|---|
| Gold | 北京 / 新年 / 音乐会 / 展现 / 经典 / 魅力 / 尉 / 健行 / 李 / 岚清 / 与 / 数千 / 首都 / 观众 / 一起 / 欣赏 |
| SLM | 北京 / 新年 / 音乐会 / 展现 / 经典 / 魅力 / 尉健行 / 李岚清 / 与 / 数千 / 首都 / 观众 / 一起 / 欣赏 |
| BERT-circular | 北京 / 新年 / 音乐 / 会 / 展现 / 经典 / 魅力 / 尉 / 健行 / 李 / 岚 / 清 / 与 / 数千 / 首 / 都 / 观众 / 一起 / 欣赏 |
| Ours | 北京 / 新年 / 音乐会 / 展现 / 经典 / 魅力 / 尉 / 健行 / 李 / 岚清 / 与 / 数千 / 首都 / 观众 / 一起 / 欣赏 |
| Gold | 北京 / 个人 / 所得税 / 增 / 二 / 成 / 五 |
| SLM | 北京 / 个人 / 所得税 / 增二 / 成五 |
| BERT-circular | 北京 / 个人 / 所得 / 税 / 增 / 二成 / 五 |
| Ours | 北京 / 个人 / 所得税 / 增二 / 成 / 五 |
| Gold | 屋内 / 炉火 / 正 / 旺 / ， / 彩灯 / 闪烁 / ， / 官兵 / 们 / 张张 / 笑脸 / 被 / 灯火 / 映 / 得 / 通红 / 。 |
| SLM | 屋内 / 炉火 / 正旺 / ， / 彩灯 / 闪烁 / ， / 官兵 / 们 / 张 / 张 / 笑 / 脸 / 被 / 灯火 / 映得 / 通红 / 。 |
| BERT-circular | 屋内 / 炉火 / 正旺 / ， / 彩灯 / 闪 / 烁 / ， / 官兵 / 们 / 张 / 张 / 笑脸 / 被灯 / 火 / 映得 / 通红 / 。 |
| Ours | 屋内 / 炉火 / 正旺 / ， / 彩灯 / 闪烁 / ， / 官兵 / 们 / 张 / 张 / 笑脸 / 被 / 灯火 / 映得 / 通红 / 。 |
| Gold | （ / 作者 / 为 / 全国 / 政协 / 副 / 主席 / 、 / 澳门 / 中华 / 总商会 / 会长 / ） |
| SLM | （ / 作者 / 为 / 全国 / 政协 / 副主席 / 、 / 澳门 / 中华 / 总商会 / 会长 / ） |
| BERT-circular | （ / 作者 / 为 / 全国 / 政协 / 副 / 主席 / 、 / 澳门 / 中华 / 总商 / 会 / 会长 / ） |
| Ours | （ / 作者 / 为 / 全国 / 政协 / 副 / 主席 / 、 / 澳门 / 中华 / 总商会 / 会长 / ） |
| Gold | 埃及 / 总统 / 穆巴拉克 / ： / 只有 / 团结 / 才 / 能 / 变 / 梦想 / 为 / 现实 |
| SLM | 埃及 / 总统 / 穆巴拉克 / ： / 只 / 有 / 团结 / 才能 / 变梦想 / 为 / 现实 |
| BERT-circular | 埃及 / 总统 / 穆 / 巴拉克 / ： / 只有 / 团结 / 才 / 能 / 变 / 梦想 / 为 / 现实 |
| Ours | 埃及 / 总统 / 穆巴拉克 / ： / 只有 / 团结 / 才 / 能 / 变 / 梦想 / 为 / 现实 |
| Gold | 药品 / 包装 / 出 / 新 / 规 / 安全 / 吃药 / 有 / 保障 |
| SLM | 药品 / 包装 / 出新规 / 安全 / 吃药 / 有保障 |
| BERT-circular | 药品 / 包装 / 出 / 新规安 / 全 / 吃 / 药 / 有 / 保障 |
| Ours | 药品 / 包装 / 出 / 新规 / 安全 / 吃药 / 有 / 保障 |
| Gold | 每逢 / 佳节 / 倍 / 思 / 廉 |
| SLM-4 | 每逢 / 佳节 / 倍 / 思廉 |
| BERT-circular | 每逢 / 佳节 / 倍 / 思廉 |
| Ours | 每逢 / 佳节 / 倍 / 思 / 廉 |
| Gold | 西沙 / 灯语 / 映 / 碧波 |
| SLM-4 | 西沙 / 灯 / 语 / 映碧 / 波 |
| BERT-circular | 西沙灯语映 / 碧波 |
| Ours | 西沙 / 灯语 / 映 / 碧波 |

Table 9: Segmentation examples.