# OpenReview forum: "Improved Unsupervised Chinese Word Segmentation Using Pre-trained Knowledge and Pseudo-labeling Transfer"
_EMNLP/2023/Conference — EMNLP 2023 Main_

### Official Review · Reviewer_CMUv · 2023-07-19

**Soundness:** 4

**Excitement:**

2: Mediocre: This paper makes marginal contributions (vs non-contemporaneous work), so I would rather not see it in the conference.

**Paper Topic And Main Contributions:**

This paper propose a new unsupervised Chinese word segmentation (UCWS) model that integrates the segmentation signal from a unsupervised segmental language model to the pre-trained BERT classifier as the pseudo-labels. The main contribution would be reducing training time (better compute efficiency) with the proposed method while providing better performance.

**Reasons To Accept:**

This paper proposed a new method for unsupervised Chinese word segmentation that provides SOTA performance with less training time.  The experimental results are relatively comprehensive and solid with reasonable analysis.

**Reasons To Reject:**

The novelty of the proposed method is limited compared with Li et al. (2023) mentioned in the paper. The proposed method is just a BERT fine-tuned on the pseudo-labels provided by a previous UCWS model, which is almost the same compared with Li et al. (2023), where the source of pseudo-labels come from a different model. Even though filtering strategy for low-confidence examples is mentioned, it is not studied in this paper.

**Reproducibility:**

5: Could easily reproduce the results.

**Reviewer Confidence:**

5: Positive that my evaluation is correct. I read the paper very carefully and I am very familiar with related work.

---

> ### Author Rebuttal · Authors · 2023-08-27
>
> We are grateful for your insightful comments on our work. We would like to address the following two issues:
>
> **Regarding the limited novelty:**
>
> Our work's novelty and contributions lie in the integration of the unsupervised Segment Language Model (SLM) and the BERT model, resulting in a much simpler training framework than Li et al. (2022) while maintaining effectiveness, as evidenced by better performance (Table 1) and faster training (Table 2). In addition, our study unveils a novel insight: the performance of Li et al. (2022) degrades significantly on words with more character lengths (in 3-gram and 4-gram) through the experiment in Figure 2, pointing out the potential weakness of Li et al. (2022) for using Perturb Masking on UCWS. Given these points, our work possesses adequate novelty and offers significant insights for future research in UCWS.
>
> **Regarding the filtering strategy issue:**
>
> While the results in Table 3 could change if we applied a filtering strategy, we chose not to include it in this work because our aim is to propose a simple yet effective framework to reduce the training time using the pre-trained BERT for UCWS (L005-L008; L047-L058). According to the results in the paper of Li et al. (2022), adding the filtering strategy with self-training contributes only 0.8\% improvement but requires much more training time. To retain simplicity and training efficiency, we did not study filtering strategies in this paper. Even so, our method still reaches state-of-the-art performance on UCWS.

---

### Official Review · Reviewer_9gJq · 2023-08-03

**Soundness:** 4

**Excitement:**

4: Strong: This paper deepens the understanding of some phenomenon or lowers the barriers to an existing research direction.

**Missing References:**

See [1] above.

**Paper Topic And Main Contributions:**

This paper proposes a simple yet effective engineering heuristic for unsupervised Chinese
word segmentation. Specifically, the proposed method first generates pseudo "answer" of
segments using an existing unsupervised segment model (LSTM-SLM), and feed them into
a BERT segmenter as the "gold" segmentation. Using early stopping, this simple heuristic
yields substantially better performance than previous baselines on UCWS, almost reaching
the theoretical upper-bound computed from inter-annotator agreements. Basically, this is a
good contribution as a short paper.

The problem is why this simple heuristic works is completely obscure. Since the output
of LSTM-SLM is fed into the BERT neural network as the "correct answer", the learned
output should be the same for the input used for training LSTM-LSM (not as shown in
Figure 1). I think that this performance improvement might come from early stopping,
namely some kind of regularization and not the model itself. Therefore, I would like to
see how the performance will vary if the training is continued without early stopping.

Since the unsupervised performance obtained in this research is quite high, it would be
an important information *where* it still fails to segment. Table 9 presents some example
outputs, but some systematic analysis would be favorable for future research in this
area.
This kind of unsupervised word segmentation is important not so much for traditional
texts, but more for colloquial texts often found in social networks. How does this method
work for such "dirty" Chinese texts with a lot of novel expressions? It cannot be
accommodated in a short paper, but is clearly an important problem for unsupervised
segmentation.

Finally, statistical unsupervised segmentation in [1] is much better than SLM-4
(for example, it achieves 82.9 F1 without using BERT). I would like to see whether the
performance will vary if the initial "true" segmentation is fairly better than LSTM-SLM.

[1] "Inducing Word and Part-of-speech with Pitman-Yor Hidden Semi-Markov Models".
Kei Uchiumi, Hiroshi Tsukahara, Daichi Mochihashi. ACL-IJCNLP 2015, pp. 1774-1782, 2015.


**Reasons To Accept:**

It provides a simple yet effective heuristic for unsupervised Chinese word segmentation.

**Reasons To Reject:**

- Why the proposed method works is completely obscure.
- The proposed method is only tested in Chinese, not for other unsegmented languages.

**Reproducibility:**

4: Could mostly reproduce the results, but there may be some variation because of sample variance or minor variations in their interpretation of the protocol or method.

**Reviewer Confidence:**

5: Positive that my evaluation is correct. I read the paper very carefully and I am very familiar with related work.

---

> ### Author Rebuttal · Authors · 2023-08-28
>
> Thank you for the praise and worthwhile comments to our work. We are happy to share our thoughts for the several addressed problems with you:
>
> **Why the proposed method works**
>
> We consider that the pre-trained BERT model (Devlin et al., 2019) plays a crucial role in our method. BERT learned profound linguistic knowledge from the massive amount of corpus during its pre-training stage. Its pre-training with masked language modeling provides opportunities for BERT to learn the word-word relationships in the pre-training corpus, and such the relationships concretely meet the goal of word segmentation tasks. In our experiments, we discover that through proper fine-tuning (e.g., early-stopping) with the pseudo-labels provided by LSTM-SLM, the BERT classifier is able to provide better segmentation results than the LSTM-SLM model itself.
>
> To further validate our approach, we will include a comparison without early-stopping in our final version.
>
> **The proposed method is only tested in Chinese**
>
> We mainly bring tests on Chinese text for the proposed method. Some researchers study word segmentation on low-resource languages like Nguni [1] with SLM, which should also work using the proposed method if there is a pre-trained language model.
>
> [1] Meyer, Francois, and Jan Buys. "Subword Segmental Language Modelling for Nguni Languages." Findings of the Association for Computational Linguistics: EMNLP 2022.
>
> **Comparison of initial segmentations**
>
> We are grateful for your suggestion to compare our work with Uchiumi et al. (2015) [2]. We have conducted additional experiments (not included in this paper) using various freely available and efficient Chinese word segmentation tools, including Jieba, SnowNLP, and HanLP. Please note that these tools may not be fully unsupervised as they might incorporate supervised methods during their construction.
>
> Using the AS dataset as an example, we found that when employing Jieba's segmentation results, which are better than those of LSTM-SLM, as pseudo-labels for the BERT-based classifier, the performance improves. The F1 score rises notably from 82.2 to 87.2. Similar improvements were observed on the other datasets and with the other CWS tools, affirming our approach’s consistency.
>
> Our findings suggest that choosing an appropriate segment module to guide the BERT-based classifier can yield substantial improvements. However, it is important to note the presence of a performance ceiling. For example, HanLP already achieves an impressive initial F1 score of 96.7 on the AS dataset. Interestingly, in this case, incorporating the BERT-based classifier does not lead to any improvement but slightly lowers the performance to 96.6.
>
> |Pseudo-labels|AS|CityU|MSR|PKU|
> |-|-|-|-|-|
> |SLM (this paper)    |79.7 &rarr; 86.4|80.3 &rarr; 85.6|79.6 &rarr; 82.4|79.6 &rarr; 84.3|
> |Jieba  |82.2 &rarr; 87.2|82.2 &rarr; 86.1|81.3 &rarr; 85.0|81.8 &rarr; 83.3|
> |SnowNLP|79.7 &rarr; 86.7|79.1 &rarr; 86.0|83.9 &rarr; 85.3|89.5 &rarr; 91.8|
> |HanLP  |96.7 &rarr; 96.6|93.8 &rarr; 94.2|88.7 &rarr; 88.5|94.8 &rarr; 94.6|
>
> (Table caption: The values at the left of the arrows are the original scores for the pseudo-labels; the values at the right are scores using our method.)
>
> [2] "Inducing Word and Part-of-speech with Pitman-Yor Hidden Semi-Markov Models". Kei Uchiumi, Hiroshi Tsukahara, Daichi Mochihashi. ACL-IJCNLP 2015, pp. 1774-1782, 2015.
>
> **More systematic error analysis**
>
> For analyzing the properties of the proposed method and the baselines, we made an analysis for different segmentation lengths in Figure 2, discovering that segmentation errors come from words with longer character lengths (3-gram and 4-gram). We appreciate your suggestion for a more systematic error analysis, which we will include in our final version.

---

### Official Review · Reviewer_sybf · 2023-08-05

**Soundness:** 3

**Excitement:**

2: Mediocre: This paper makes marginal contributions (vs non-contemporaneous work), so I would rather not see it in the conference.

**Paper Topic And Main Contributions:**

Previous work on unsupervised Chinese word segmentation incorporates linguistic knowledge from pre-trained language models using parameter-free probing techniques. The authors found that the approaches suffer from increased training time due to multiple inference pass using pre-trained LM.

The paper proposes to combine a segment language model and a standard sequence labelling model with PLM.
Segment Language model can provide segmentation signal. With the samples, pretrained BERT classifier can be trained under a pseudo-labeling framework.

Experiments show that the approach achieves state-of-the-art performance on the eight UCWS tasks and reduces the training time compared to the previous work.

**Reasons To Accept:**

a) The paper is well written. The motivation and methods are easy to understand.

b) The result looks good and achieves SOTA on the eight test sets.

**Reasons To Reject:**

There is some problem with experiment setting if I understand correctly. The test set should be blind during the training process. As described in A.2 section in Appendix (L413-418), SLM would infer on test set to get pseudo-labels. And the BERT classifier is trained on the pseudo-label of test set directly.

It is unfair to compare with the previous work as test set (including the raw text without any label) should not be provided to the models during training process. If test set is used when training, the model can bias towards sentence on test set.
It is better to try the setting where the classifier is trained over training samples with pseudo-labels.

---
Update:
According to the rebuttal, Li et al., 2022 also used test set without labels (raw test data). Seems that SLM-4 and MSLM do not uses this information. The paper should label which method use raw test data in Table 1 for better understanding.

I raise the soundness score according to the rebuttal.

**Reproducibility:**

4: Could mostly reproduce the results, but there may be some variation because of sample variance or minor variations in their interpretation of the protocol or method.

**Reviewer Confidence:**

3: Pretty sure, but there's a chance I missed something. Although I have a good feel for this area in general, I did not carefully check the paper's details, e.g., the math, experimental design, or novelty.

---

> ### Author Rebuttal · Authors · 2023-08-27
>
> Thank you for your constructive comments and for recognizing the contributions of our work. We would like to specifically address your concern regarding the use of the test set during training.
>
> You are correct that the test sets without the ground-truth labels were used to infer pseudo-labels through our Segment Language Model (SLM), as we described in Appendix (L413-418). We'd like to emphasize that these pseudo-labels are not derived from actual test labels but are generated by the SLM alone, without any ground-truth information.
>
> However, Bert-Circular (Li et al., 2022) also used examples from test sets with inferred pseudo-labels to train their BERT model. Although Li et al. (2022) did not explicitly mention this setting in their paper, the supported evidence can be found via their publicly available code on GitHub. As a result, the comparisons made in our paper are fair.
>
> We understand the potential concern about bias towards the test set. However, the goal was to demonstrate the efficiency of our approach with existing methods under the same conditions. We are willing to provide further comparisons using different training sources if our paper gets accepted.

---

### Meta-Review · Area_Chair_ggPC · 2023-09-19

**Recommendation:** 4

**Metareview:**

This paper explores improving unsupervised Chinese word segmentation by fine-tuning BERT to this task with pseudo-labels generated by an unsupervised segment model. The authors report that their approach yields improvements over prior state of the art on seven out of eight CWS datasets, while requiring significantly less training time.

The reviewers agree that the approach is simple, sound, competitive with prior work while requiring less time to train. The reviewers' initial concerns about how the work is distinct from or comparable to prior work seem to have been resolved with the author response. The most critical reviewer assigned strong soundness and excitement scores, which suggests that their criticisms were meant to be taken as suggestions for improvement rather than as reasons to reject.

---

### Decision · Program_Chairs · 2023-10-07

**Decision:**

Accept-Main

**Comment:**

This paper explores improving unsupervised Chinese word segmentation by fine-tuning BERT to this task with pseudo-labels generated by an unsupervised segment model. The authors report that their approach yields improvements over prior state of the art on seven out of eight CWS datasets, while requiring significantly less training time.

The reviewers agree that the approach is simple, sound, competitive with prior work while requiring less time to train. The reviewers' initial concerns about how the work is distinct from or comparable to prior work seem to have been resolved with the author response. The most critical reviewer assigned strong soundness and excitement scores, which suggests that their criticisms were meant to be taken as suggestions for improvement rather than as reasons to reject.